# Have environmental regulations promoted green technological innovation in cities? Evidence from China's green patents

**Ming Zhang[1], Wancheng Xie[1]\*, Wen Gao[2]**

**1** School of Economics and Business Administration, Chongqing University, Chongqing, China, **2** School of Banking and Finance, University of International Business and Economics, Beijing, China

\* xiewancheng@cqu.edu.cn

**Data Availability Statement:** All relevant data are within the paper and its Supporting information files.

## Abstract

Under the background of global response to climate crisis and environmental pollution, environmental regulation plays an increasingly important role in green technology innovation. This paper uses data from 280 Chinese cities from 2003 to 2019 to empirically answer the question whether environmental regulation can improve the level of urban green technology innovation. It is found that environmental regulation has a significant positive effect on green technology innovation. Under the heterogeneity of economic geographical region and resource-based city, environmental regulation has positive promoting effect on urban green technology innovation. Heterogeneity results also show that environmental regulation significantly promotes green technology innovation in central and resource-based cities, but does not significantly promote green technology innovation in eastern and western cities and non-resource-based cities. Further research shows that environmental regulation can promote the level of green technology innovation through the two transmission mechanisms of government technology input and foreign direct investment. For the Chinese local government which is implementing the policy of green economic transformation, the formulation of scientific regional environmental policy is beneficial to improve the level of green technology innovation, increase government technology input and optimize the foreign investment environment.

## 1. Introduction

Over the past 40 years of reform and opening up, China's economy has achieved leapfrog development and has become the world's second largest economy. The rapid economic development has also brought serious ecological and environmental problems. On the one hand, rapid urbanization has put pressure on the environment. On the other hand, China, as a world factory, has produced serious industrial pollution emissions [1, 2]. According to the investigation [3, 4], the cost of environmental degradation in 2013 increased by 301% compared with 2004, accounting for 3.3% of the annual GDP. The Asian Development Bank and the Chinese government jointly conducted a survey and research on the environment during China's Twelfth Five-Year Plan. The report revealed that less than 1% of China's largest 500 cities meet

**Funding:** "Social Science (17YJC790024). The project provided only financial support for this study. Meanwhile, the funders had no role in study design, data collection and analysis, decision to publish, or preparation of the manuscript."

**Competing interests:** NO authors have competing interests.

the air quality recommended by the World Health Organization. According to standards, among the 10 most polluted cities in the world, 7 are in China [5]. These facts show that China continues to pay a huge economic cost for environmental damage, and also reflect the increasing pressure of environmental pollution control.

In order to curb the deteriorating trend of ecological and environmental problems, the Chinese government has successively promulgated environmental laws and regulations on air pollution, water pollution, and soil pollution since 2008. "Green waters and green mountains are the golden mountains and silver mountains" have been set as the basic national policy of the country's development. Strict legal systems and regulations have improved China's environmental problems [6–8]. According to the Swiss IQAir report [9], only 2 Chinese cities were among the 20 most polluted cities in the world in 2019, and Beijing was listed among the 200 most polluted cities in the world, which fully shows that China's environmental policies have achieved good results. Although the Chinese government has made great achievements in environmental governance, the environmental problems faced by Chinese cities are still very serious. In 2019, 47 cities in China are located in the 100 most polluted cities in the world, and only 2% of the cities have annual pollution. The average PM2.5 concentration is less than the World Health Organization standard value [9], and environmental governance is still an urgent issue for sustainable development in Chinese cities.

Environmental governance not only requires the government to issue relevant environmental regulations to restrain market environmental pollution, but also requires market subjects to choose clean technologies for production and business activities [10]. Especially in the context of more and more strict environmental policies and regulations: on the one hand, enterprises bear higher and higher environmental taxes and environmental penalties; On the other hand, the environmental threshold for enterprises to enter the market is getting higher and higher. This requires enterprises to promote green technology innovation to deal with the cost of environmental pollution in the long run. Green technology innovation is considered as an important means to improve ecological damage and achieve sustainable development [11, 12], but the academic circle has not reached a unified conclusion on the relationship between environmental regulation and technological innovation. Porter's hypothesis shows different effects in different countries and regions, and the influence mechanism of environmental regulation on green technological innovation is not clear [13, 14]. Compared with non-green technology innovation, is the impact of environmental regulation on green technology in Porter hypothesis valid? What is the mechanism by which environmental regulation affects green technology innovation? What factors will cause the heterogeneous impact of environmental regulation on green technology innovation? In order to answer the above questions, we studied the impact of environmental regulation on green technology innovation in 283 prefecture-level cities in China.

The marginal contributions of this paper are as follows: First, based on green patent data, this paper answers the question of the impact of environmental regulation on green technology innovation at the city level. Compared with previous studies using technological innovation and total factor productivity to express green technology innovation [15–17]. Second, this paper incorporates foreign direct investment and government technical input into the analysis framework of the impact of environmental regulation on green technology innovation. Compared with previous research on financial support and low-carbon policies [16, 18]. Third, this paper analyzes the moderating effect of differences in geographic regions and resource endowments on the impact of environmental regulation on green technology innovation. Compared with previous studies that only focused on geographical differences and ignored resource endowments [17], the conclusion of this paper not only enriches the heterogeneity research on the impact of environmental regulation on green technology innovation, but also provides a

path reference for cities with different resource endowments to promote green technology innovation.

The structure of this paper is as follows: The second chapter is literature review and theoretical hypothesis. The third chapter is model setting and data description. The fourth chapter is the basic model regression, robustness test, heterogeneity analysis and mediating effect test. The fifth chapter is the conclusion and policy suggestion.

## 2. Literature review and theoretical hypotheses

### 2.1 Literature review

Technological innovation is a process of constantly upgrading and transforming existing technologies [19]. Antweiler et al. [20] called technological effect a key factor in dealing with environmental pollution, and this demonstration quickly aroused scholars' research on technological innovation and environmental pollution control. However, not all technological innovations are beneficial to the environment. Green technological innovation is distinguished from non-green technological innovation, and is considered to be a technology that can improve environmental pollution and improve environmental governance [21, 22]. OECD [23] defined green innovation as new or improved product development, organizational structure, production process, sales methods and other actions to reduce environmental pollution. Therefore, scholars generally define green technology innovation as the collection of technologies, systems and products that promote environmental governance and ecological protection and achieve sustainable development [24]. Green technology innovation has become an important technological means of industrial green transformation and environmental governance [22, 25], which plays a crucial role in urban sustainable development [12, 26]. However, scholars hold different opinions on the measurement of green technological innovation. Some scholars disintegrate green technological innovation into green product innovation and green process innovation, and refer to green product innovation by sales of new products per unit energy consumption in the calculation process. Green process innovation is referred to by the sum of internal expenditure of R&D expenditure and technological transformation [27–29]. However, Tong et al. [30] pointed out that the number of patents refers to technological innovation more scientifically than R&D.

Environmental regulation is considered as an important means for the government to regulate the green development of market entities [31, 32]. After the Chipko movement in India, scholars have a new understanding of the definition of environmental regulation. Existing studies generally define environmental regulation as all kinds of tangible and intangible institutional constraints for the purpose of ecological protection and environmental governance. Research on the impact of environmental regulation on green technology innovation has attracted extensive attention from academic researchers. The existing literature on the relationship between environmental regulation and green technology innovation is mainly divided into three categories.

Scholars who hold the first view mainly focus on porter's hypothesis to study the promotion effect of environmental regulation on green technology innovation. Porter hypothesis holds that appropriate environmental regulation will stimulate enterprise technological innovation to improve enterprise productivity, and then strengthen enterprise technological innovation [33, 34]. Based on the analysis of technology bias model, Acemoglu [10] concluded that environmental regulation can guarantee economic growth while realizing enterprise green technology innovation. In addition to the relevant environmental policies formulated by the government, using economic means to punish polluters and illegal enterprises is more conducive to stimulating green innovation of enterprises [35, 36]. In addition, to ensure the effective

implementation of environmental policies, the government has also introduced industrial, fiscal and financial policies to support green innovation of enterprises. These policies not only limit enterprises' pollution emission behavior, but also reduce the cost of enterprises' green technology innovation and further enhance the motivation of enterprises' green technology innovation [37–39]. Scholars who hold the second view oppose the assertion of Porter's hypothesis. They believe that Porter's hypothesis ignores the heterogeneity of firms and environmental regulation may have a negative impact on firms' technological innovation.

For some enterprises with high cost of green technology innovation, if environmental regulations restrict some technical standards, they may have no incentive to choose green technology innovation [13]. For example, Shi et al. found that China's carbon emission environment pilot policy significantly inhibited enterprise innovation and reduced enterprise productivity [40]. At the same time, the impact of environmental regulation can not ignore the double externality of green technology innovation. On the one hand, commercial banks and other institutions generally do not believe that green technology innovation can improve enterprise performance, so the financial market does not support green innovation. On the other hand, due to the externality of environmental pollution itself, enterprises are not willing to bear the cost of the public sector, so as to improve their own green technology innovation [41, 42].

Scholars who hold the third view believe that the impact of environmental regulation on green technology innovation cannot be determined. For example, Guo et al. decomposed green technology innovation into green product innovation and green process innovation, and found that there is a U-shaped relationship between environmental regulation and green technology innovation [29]. Li et al. also verified this view by using the data of China's construction industry [43]. They believed that the impact of environmental regulation on green technology innovation in China's construction industry was nonlinear. In addition, some scholars oppose Porter's hypothesis and believe that there is no relationship between environmental regulation and green innovation behavior. For example, Brunnermeierab and Cohen found that the increase of environmental policies did not lead to additional innovation of enterprises [44]. Yuan and Xiang used China's cleaner production standards to conduct research and found that environmental regulations only significantly improved the profitability of enterprises, but did not promote enterprise innovation [45].

It can be seen that the impact of environmental regulation on green technology innovation is uncertain, which may be determined by sample deviation, regional difference and enterprise difference, etc. [46, 47]. At present, there are limited studies on the impact of environmental regulations on the mechanism of green technology innovation, and most of the literatures are mainly analyzed from the perspective of enterprises, such as green finance, fiscal subsidies and other factors. Some literatures also studied the mechanism of environmental regulation on green technology innovation at the national and provincial levels, such as industrial structure and foreign investment. Wu et al. analyzed the impact of fiscal subsidies on enterprises' green technology innovation by using the data of Chinese listed companies, and found that fiscal subsidies improved the efficiency of enterprises' green technology innovation [48]. Hong et al. based on panel data of Chinese listed companies, finds that green credit is beneficial to green technology innovation of state-owned enterprises [49]. Some literatures also studied the impact of foreign direct investment and government investment in technological innovation on green technology innovation. For example, Wang et al. found that the inflow of FDI significantly improved regional ecological efficiency [50]. Hao et al. made use of Chinese provincial data and found that reverse technology spillovers of OFDI improved the technological innovation level of the home country [51]. Guo et al. found that government research funding has a positive effect on enterprises' green technology innovation [29]. Li et al. also found that macro policies affect the level of green development through R&D. Of course, some literatures focus

on the relationship between environmental regulations and foreign direct investment [52]. Scholars debate whether environmental regulations are "pollution refuge" or "pollution halo" for FDI [53], there is no consistent conclusion so far. Therefore, there is still a lack of mechanism research on environmental regulation and green technology innovation.

## 2.2 Theoretical hypotheses

**2.2.1 Base hypothesis.** China's urban environmental regulation mainly manages and restrains market subjects through government actions, such as sewage charges, environmental taxes, environmental permits, administrative penalties and other means. As early as the 1980s, the Chinese government began to implement market-based environmental incentives such as sewage charges, sulfur dioxide charges, and "three simultaneous" deposits. Under China's "environmental vertical" governance system, local officials generally enforce stricter environmental measures because of "environmental goals" and "promotion pressure". These strict environmental policies force enterprises to carry out green technology innovation to avoid the risk of being suspended, penalized or shut down [54–56]. According to the porter hypothesis theory, market main body of green innovation level by "innovation" and "compliance costs" effect, conform to the requirements of the development of environmental policy will not become the "drag" of the enterprise. On the contrary, these policies will not only make enterprises can enhance the level of green innovation and will also improve enterprise beneficial position in the market [33, 57]. On the one hand, high-intensity environmental regulations require enterprises to reduce pollution emissions by improving the level of green technology innovation, so as to improve production technology to increase productivity and enhance technological advantages to compensate for the cost lost in the process of technological innovation, forming the "innovation compensation" effect [58]. On the other hand, environmental regulations force enterprises to consider the cost of green innovation, especially in the context of stricter environmental standards. Under the principle of minimizing "following costs", enterprises may invest money to pay pollution charges in the short term. But in the long run, in order to reduce pollution emissions in the future production process, enterprises have to increase investment in green innovation [59]. Heterogeneity also influences the relationship between environmental regulation and green technology innovation. For example, the institutional environment and economic development level of eastern China are slightly higher than that of central and western China due to the different economic development level caused by geographical location. This results in differences in environmental policies made by local governments, resulting in "pollution transfer", which further reduces the local level of green technology innovation [60–62]. Resource attributes determine that resource-based cities have more energy-consuming and polluting industries than non-resource-based cities, such as mining, oil smelting, coal industry and metal smelting. Under the condition that environmental policies tend to be strict, polluting industries need to improve green innovation level more than clean industries [10]. Therefore, we have hypothesis 1 and hypothesis 2.

Hypothesis 1: Environmental regulation has a positive effect on green technology innovation.

Hypothesis 2: Different economic regions and urban resource types may cause the heterogeneous impact of environmental regulation on green technology innovation.

**2.2.2 Mechanism hypothesis.** The "pollution refuge" hypothesis points out that FDI has a negative impact on the environment of the host country, which has been demonstrated by many scholars. For example, Antweilier et al. found that loose environmental regulations make polluting enterprises transfer from rich countries to poor countries [20]. Candau and

Dienesch found that polluters are more likely to choose countries with looser environmental standards [63]. However, some scholars believe that strict environmental regulations limit FDI in polluting industries and are conducive to attracting FDI in high-tech and green industries. FDI brings advanced production equipment, advanced technology and advanced management experience to developing countries, which improves the environmental quality of developing countries or regions [20, 64]. Zhang and Zhou used China's provincial panel data to study and found that FDI contributes to China's energy conservation and emission reduction [53]. Earnhart et al. found that the host country's FDI attraction was influenced by its own environmental regulation and information disclosure. The countries with more ISO14001 certifications attracted more FDI [65]. It can be seen that the strength of environmental regulations affects the inflow of foreign direct investment [37]. China's environmental policies formulated since the end of the 20th century have achieved certain results [66], which provides a basis for attracting clean foreign direct investment. Foreign direct investment not only brings green production technology and pollution treatment technology, but also improves enterprises' ability of green management and organization, helps local enterprises learn advanced green production technology and management experience, as well as helps host countries upgrade old technology from traditional enterprises to modern clean technology. Thus, the level of green technology innovation of enterprises is improved [67, 68]. So we propose hypothesis 3.

Hypothesis 3: Foreign direct investment has a mediating effect between environmental regulation and green technology innovation. Environmental regulation improves the level of green technology innovation through increasing foreign direct investment.

From the enterprise level, green technology innovation is characterized by high investment, high risk and long time, and enterprises' choice to engage in green technology innovation depends on their operating profits [69]. When the "innovation compensation" income brought by green technology innovation is greater than the pollution cost, Enterprises are more active in green technology innovation activities [10]. When enterprises are strongly regulated by the local government, on the one hand, they have to increase R&D investment in the field of green innovation in order to avoid high pollution charges, thus promoting the improvement of green innovation level of enterprises. On the other hand, enterprises are under dual pressure from the public and investors. Tang and Tang found that the public tends to commercialized media to urge enterprises to provide solutions for their pollution [70]. Xu et al. (2016) found that investors tend to give lower valuations to companies with poor environmental performance, while giving higher valuations to companies with good environmental performance [71]. Therefore, enterprises are under external pressure to increase investment in technological innovation and promote green technological innovation so as to achieve good environmental performance. From the perspective of local governments, in order to implement environmental policies and give full play to the effects of environmental policies, local governments generally adopt environmental subsidies to alleviate the lack of funds for enterprises' green innovation research and development, so as to stimulate the improvement of enterprises' green technology innovation level [48, 72, 73]. At the same time, in order to encourage and support green innovation by market subjects, the government urges financial institutions to provide green bonds and green loans specifically for green innovation and enhance financial support for green innovation [74]. In addition, under the promotion pressure of "environmental protection goals", Chinese officials generally take strong administrative measures to punish polluters, which in turn forces enterprises to increase their investment in green innovation R&D [55]. Therefore, we believe that the implementation of environmental regulation increases the investment of green technology innovation in the market, and then improves the level of green technology innovation in enterprises. We came up with hypothesis 4.

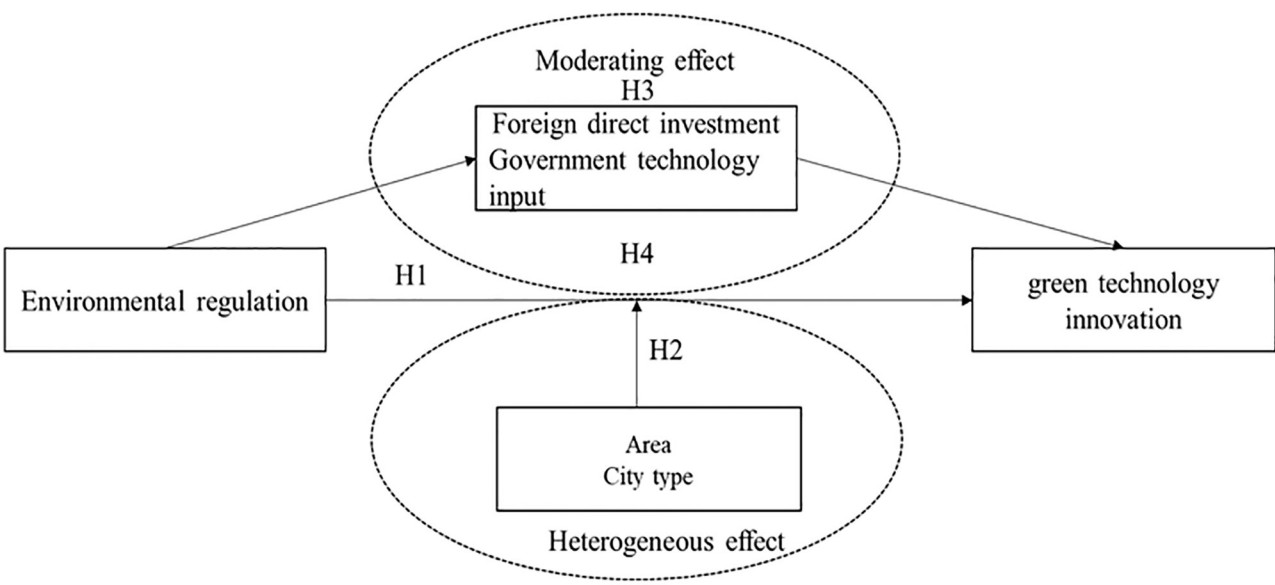

**Fig 1. The mechanism of environmental regulation on green technology innovation.**

Hypothesis 4: government technology input has a mediating effect between environmental regulation and green technology innovation, and environmental regulation improves the level of green technology innovation through government technology input.

According to the above analysis, the mechanism of environmental regulation on green technology innovation in resource-based cities is proposed, as shown in Fig 1:

## 3. Methodology and variables

### 3.1 Methodology

In order to explore the impact of environmental regulation on urban green technology innovation, the benchmark panel model constructed in this study is as follows. At the same time, due to the large GTI among different regions, this paper also controls the individual effects of city and time.

$$\text{GTI}_{it} = \alpha_0 + \alpha_1 ER_{it} + \alpha_2 Control_{it} + City_i + Year_t + \varepsilon_{it} \qquad (1)$$

Where $i$ and $t$ represent city and year respectively. $\text{GTI}_{it}$ is the dependent variable denotes the green technology innovation. $ER_{it}$ is the independent variable which denotes the environmental regulation. $Control_{it}$ is the control variable matrix, including Region per capita GDP (ED), industrial structure (IS), resource endowment (RE), government intervention (GI), education level (EL) and infrastructure construction (TIC). $City_i$ represents city fixed effects, $Year_t$ represents time fixed effects, and $\varepsilon_{it}$ represents a disturbance term.

According to the previous analysis, the impact of environmental regulations on green technological innovation not only has a direct path, but also indirectly affects green technological innovation through the transmission mechanism of government technology input. This paper refers to the model of Baron and Kenny on mediating effect to test Hypothesis 3 [75]. The following model is set:

$$GOT_{it} = \beta_0 + \beta_1 ER_{it} + \beta_2 Control_{it} + City_i + Year_t + \varepsilon_{it} \qquad (2)$$

$$\text{GTI}_{it} = \delta_0 + \delta_1 ER_{it} + \delta_2 GOT_{it} + \delta_3 Control_{it} + City_i + Year_t + \varepsilon_{it} \tag{3}$$

Where $GOT_{it}$ represents regional government technology input which is the mediating variable and other settings are the same as Eq (1). This present study uses the stepwise regression method to conduct the intermediary test. The test procedure is as follows: The first step is to regress Eq (1). If $\alpha_1$ is significant, which indicates that the overall effect of the strengthening of environmental regulations on green technological innovation exists, and the next test is carried out. Otherwise, the mediation effect does not exist and the test is terminated. The second step is to regress Eq (2) to test the impact of environmental regulations on government technology input. The third step is to regress Eq (3) to test the direct effect of environmental regulation on green technological innovation and the mediating effect of investment in technological innovation. If $\beta_1$ and $\delta_2$ are both significant, which means that the indirect effect is significant, and the fourth step test is performed. The fourth step is to compare the signs of $\beta_1 \times \delta_2$ and $\delta_1$. If the signs are the same, which means that there is a mediating effect. If the signs are different, there is no mediating effect.

In addition, this present research also to explore environmental regulations have an indirect impact on green technological innovation through foreign direct investment. In the same way, this present research adopts the mediating effect model to test hypothesis 4.

$$FDI_{it} = \gamma_0 + \gamma_1 ER_{it} + \gamma_2 Control_{it} + City_i + Year_t + \varepsilon_{it} \tag{4}$$

$$\text{GTI}_{it} = \varphi_0 + \varphi_1 ER_{it} + \varphi_2 FDI_{it} + \varphi_3 Control_{it} + City_i + Year_t + \varepsilon_{it} \tag{5}$$

Where $FDI_{it}$ represents foreign direct investment which is the Mediating variable and other settings are the same as Eq (1). Meanwhile, the inspection steps are the same as above.

## 3.2. Variable and definition

**3.2.1. Dependent variable.** Green Technological Innovation (GTI). Green technological innovation is one of the important factors to realize the green development of cities. In general, the existing literature related to the measurement of GTI can mainly be divided into three parts: First, measuring technological innovation from the perspective of technological innovation investment, scholars mainly adopt R&D expenditure [76]. Second, constructing a comprehensive index to evaluate and measure technological innovation: such as green total factor productivity [77]. In addition, measured from the perspective of technological innovation output: such as green patent based on "IPC Green Inventory" proposed by WIPO [78, 79]. However, due to data limitations, R&D investment cannot effectively measure green technological innovation. Meanwhile, when R&D investment is inefficient, green technological innovation may be overestimated [80]. Patents filed in Green technologies are a relevant indicator for approximating environmental innovations [81]. As the number of patent applications in some cities is 0, this study adopts the number of green patent applications plus 1 followed by logarithm to represent GTI.

**3.2.2. Independent variable.** Environmental regulation (ER). Environmental regulation means that the administrative department formulates environmental-related laws, regulations and standards, directly restricts the pollution discharge of enterprises, so as to ultimately achieve the goal of improving the ecological environment. The existing environmental regulation measurement methods mainly include: The first is the single index method, which is measured by the number of environmental regulation policies and regulations issued, the amount of environmental pollution control investment, the proportion of pollution control investment

in the cost or output value of the enterprise, and the collection Sewage charges etc [82, 83]. The second is the comprehensive index method, which uses the entropy weight method to weight each single indicator, such as the area's pollution discharge or wastewater discharge compliance rate, the removal rate of sulfur dioxide, the removal rate of soot and dust, and the comprehensive utilization rate of solid waste. Since the amount of industrial wastewater discharge will not be announced after 2010, this article draws on the research methods of Huang et al. using industrial waste water discharged, industrial sulfur dioxide emission, industrial soot(dust) emission, industrial solid waste comprehensive utilization rate and urban domestic sewage treatment rate [84]. Then, the entropy method is used to measure ER. The advantage of using this method is that it can reflect the implementation of environmental regulations more comprehensively and accurately. Among them, the larger the environmental regulation value, the stronger the implementation of local environmental regulation.

**3.2.3. Mediating variables.** Foreign direct investment (FDI). The pollution refuge hypothesis believes that loose environmental regulations will attract an increase in foreign direct investment [85]. Meanwhile, FDI can bring about changes in the host country's technology, industrial structure, and market size. These factors have a certain impact on regional green technology innovation and have an impact on the regional ecological efficiency [50]. In this present study, In this paper, FDI is expressed as the logarithm of the amount of foreign capital investment actually utilized.

Government technology input (GOT). In this paper, the government's subsidies for promoting enterprises' technological innovation and R&D through direct and indirect means are collectively referred to as government technology input. The government's technological innovation R&D subsidy to enterprises is one of the important factors to promote regional technological innovation [86]. In order to avoid the negative impact of environmental regulations on regional green technological innovation, the government often tries to stimulate the industry through subsidies and other means [87]. Considering that there is no specific statistics on government technology subsidies at the city level. However, science and technology expenditures are mainly science and technology expenditure items in public budget expenditures, which can better reflect the government's R&D subsidies to enterprises. Therefore, this paper uses the ratio of science and technology expenditure to GDP to represent government technology input.

**3.2.4. Control variables.** Economic development (ED). The EKC hypothesis holds that there is an inverted u-shaped relationship between environmental pollution and the level of economic development [88]. Economic development will lead to an increase in environmental pollution, but with economic development, the opportunity cost of health and the environment will increase until people attach importance to environmental protection. This present study selects the logarithm of per capita GDP to represent regional economic development. Industrial structure (IS). Environmental protection and green technology innovation are closely related to industrialization. In the early stages of industrialization, people paid more attention to the improvement of production technology, while ignoring environmental protection. With the increase of ecological pressure and entering the period of industrial transformation, environmental protection has received widespread attention and the industrial structure has changed accordingly [89]. This present study selects ratio of the output value of secondary industry to regional GDP to represent the regional industrial structure. Resource endowments (RE). Regions with sufficient resource endowments may be overly dependent on natural resources and fall into the "resource curse" trap, affecting the local green technology innovation and development [90]. This present study uses Ratio of the number of people in the extractive industry to the total number of employees to represent. Government intervention (GI). As China continues to promote green development, the government will adopt relevant

laws, regulations and subsidies to improve resource efficiency and alleviate environmental pollution and other issues [91]. So as to help improve the level of green technology innovation. This present study uses ratio of public finance expenditure to regional GDP to express. Education level (EL). Drawing lessons from [92], combined with China's setting of education years, the education years corresponding to the education levels of elementary school, junior high school, and high school are set to 6 years, 3 years, and 3 years, respectively. Therefore, we use the number of elementary school students in the area multiplied by 6, the number of middle school students multiplied by 6, the number of high school students multiplied by the sum of 12 and the logarithm of the comprehensive index is used to measure the education level. Transportation infrastructure construction (TIC). Transportation infrastructure construction is generally represented by traffic density and total freight volume. As the total freight volume is used as a flow indicator, it can drive the flow of resources and labor, inject "vigour" into the development of local green technology innovation. So as to more intuitively reflect the impact of transportation infrastructure on green technology innovation. Therefore, this present study uses the total freight volume to represent the transportation infrastructure construction. Table 1 presents the definitions of the variables selected in this paper.

### 3.3. Data and descriptive statistics

Considering the pertinence of the research, continuity of data availability, and comparability between cities, this study selects the data of 280 cities in China from 2003 to 2019, with 4760 observations. We derive the related variables from the China City Statistical Yearbook and the China Environment Statistical Yearbook. Meanwhile, all the datasets of the study sources from the Economy Prediction System (EPS) database (The EPS data platform has built a series of professional databases, including: China Industry Business Performance Data, the China Environment Statistical Yearbook, etc. Available online at: http://olaptest.epsnet.com.cn/). Furthermore, this study uses the interpolation method to complete some missing values in the variables. Table 2 gives a statistical description of the main variables.

## 4. Empirical results and discussion

### 4.1. Basic model regression results

Columns (1)-(7) of Table 3 report the regression results between environmental regulation and firm eco-innovation. According to the statistics of Hausmann's chi-square test, we rejected

**Table 1. Variables definition.**

| Variable classification | Variable symbol | Variable definitions |
|---|---|---|
| Dependent variable | GTI | The green patent application count plus 1 followed by logarithm |
| Independent variable | ER | Comprehensive score of environmental regulation |
| Mediating variables | FDI | The logarithm of The amount of foreign capital actually utilized |
| | GOT | Ratio of science and technology investment to public finance expenditure |
| Control variables | ED | Region per capita GDP, take the natural logarithm |
| | IS | Ratio of the output value of secondary industry to regional GDP |
| | RE | Ratio of the number of people in the extractive industry to the total number of employees |
| | GI | Ratio of public finance expenditure to regional GDP |
| | EL | Comprehensive index and take the logarithm |
| | TIC | the total freight volume, take the natural logarithm |

**Table 2. Descriptive statistics.**

| Variable | N | Mean | St. dev | Min | Max |
|---|---|---|---|---|---|
| GTI | 4760 | 2.919 | 1.815 | 0.000 | 8.990 |
| ER | 4760 | 0.714 | 0.182 | 0.132 | 0.995 |
| FDI | 4760 | 9.439 | 2.059 | 0.000 | 14.152 |
| GOT | 4760 | 0.002 | 0.003 | 0.000 | 0.063 |
| ED | 4760 | 10.210 | 0.850 | 4.595 | 15.675 |
| IS | 4760 | 0.476 | 0.111 | 0.090 | 0.910 |
| RE | 4760 | 0.055 | 0.093 | 0.000 | 0.581 |
| GI | 4760 | 0.192 | 0.207 | 0.015 | 6.041 |
| EL | 4760 | 15.150 | 0.756 | 11.918 | 18.506 |
| TIC | 4760 | 7.382 | 2.100 | -5.221 | 11.820 |

the random effect model setting at the 1% level and chose the fixed effect model. Meanwhile, in order to describe the influence of addition of the control variables on the regression results, the model is regressed by adding the control variables step-by-step. The regression results of the model (Table 3) show that ER has passed the significance test in the models of different control variables, and was positively correlated with green technological innovation (GTI). We can find from column (7) that, after considering all control variables and fixed effects, the regression coefficients between ER and GTI is 0.305 and the correlations between them are significantly positive at the 5% level. This shows that higher levels of environmental regulation can during the study period contributed to the improvement of the GTI level, which validates H1. This conclusion is consistent with the Porter's hypothesis that environmental regulations are conducive to promoting innovation (Porter and Van der Linde, 1991). This also shows that properly designed ER and the implementation of a series of energy saving and emission reduction policies are conducive to the development of green ecological innovation in China [77].

In addition, from the perspective of the other control variables, the following results are found from column (7). ①RE will significantly promote the improvement of the GTI. This

**Table 3. Baseline regression.**

| Variables | GTI | | | | | | |
|---|---|---|---|---|---|---|---|
| | (1) | (2) | (3) | (4) | (5) | (6) | (7) |
| ER | 0.348** (2.23) | 0.371** (2.39) | 0.339** (2.22) | 0.325** (2.2) | 0.312** (2.15) | 0.314** (2.26) | 0.305** (2.21) |
| RE | | 2.170*** (2.77) | 1.999 ** (2.56) | 1.982** (2.54) | 2.012*** (2.59) | 1.871** (2.48) | 2.025** (2.66) |
| ED | | | 0.168** (2.07) | 0.140* (1.69) | 0.132 (1.65) | 0.157** (2.1) | 0.148** (1.98) |
| IS | | | | 0.250 (0.75) | 0.271 (0.81) | 0.503 (1.57) | 0.453 (1.41) |
| GI | | | | | -0.305*** (-4.84) | -0.203*** (-3.68) | -0.196*** (-3.55) |
| EL | | | | | | 0.662*** | 0.651*** |
| | | | | | | (4.35) | (4.30) |
| TIC | | | | | | | 0.027** (2.44) |
| Cons | 1.090*** (11.73) | 0.952*** (9.42) | -0.551 (-0.75) | -0.396 (-0.56) | -0.295 (-0.43) | -10.71*** (-4.4) | -10.65*** (-4.4) |
| City FE | YES | YES | YES | YES | YES | YES | YES |
| Year FE | YES | YES | YES | YES | YES | YES | YES |
| R² | 0.347 | 0.299 | 0.350 | 0.344 | 0.355 | 0.573 | 0.574 |
| N | 4760 | 4760 | 4760 | 4760 | 4760 | 4760 | 4760 |

Note: The values of t are given in brackets.

***, **, * represent the different significance levels (1%, 5%, 10% respectively).

may be due to the Chinese government's pursuit of green development goals in recent years, especially for resource-based cities, increasing investment in green innovation and optimizing industrial and energy utilization structures [93]. ②The influence of ED on the GTI is significantly positive, indicating that indicating that the higher the degree of economic development, the more conducive it is to promoting green technological innovation [47, 94]. ③The influence of IS on the GTI is significantly positive, but it does not pass the significance test. ④GI will significantly inhibit the GTFP, indicating that the efficiency of China's government fiscal support still needs to be improved at this stage. Meanwhile, extensive investment and "one-size-fits-all" fiscal support models should be avoided. ⑤EL has a significant and promoting effect on GTI, mainly because the large number of talents produced in China in recent years and the high rate of achievement conversion and technological contribution rate. ⑥TIC also has a significant positive effect on GTI. It indicates that the improvement of the level of transportation infrastructure construction is conducive to the promotion of green technological innovation of cities.

## 4.2. Robustness test

To make our results more credible, we have conducted a large number of robustness tests through a variety of methods. The results are shown in Table 4. First, as China's prefecture-level cities or provincial capital cities are the facades of this region, the effects of any national policies must be first shown in provincial capital cities. This has also prompted the provincial government to concentrate more political and economic resources in the provincial capital to ensure that the provincial capital can fulfil the relevant national policy requirements. If this kind of government behavior is not eliminated, it will cause an overestimation of the effect of ER on green technological innovation. Therefore, we exclude all provincial capital cities record the dependent variable as D-GTI, and the regression results are reported in column (1). We

**Table 4. Robustness test.**

| Variables | D-GTI | R-GTI | GTI | GTI | GTI |
|---|---|---|---|---|---|
| | **(1)** | **(2)** | **(3)** | **(4)** | **(5)** |
| ER | 0.341** | 0.189** | | | 0.279*** |
| | (2.36) | (2.09) | | | (3.21) |
| R-ER | | | 0.277*** | | |
| | | | (2.64) | | |
| L.ER | | | | 0.159* | |
| | | | | (1.77) | |
| Control variables | YES | YES | YES | YES | YES |
| Constant | -10.14*** | -13.21*** | -10.56*** | -10.16*** | -5.83*** |
| | (-3.85) | (-13.08) | (-4.38) | (-9.98) | (-7.51) |
| City FE | YES | YES | YES | YES | YES |
| Year FE | YES | YES | YES | YES | YES |
| Adjusted $R^2$ | 0.559 | 0.494 | 0.576 | 0.556 | |
| AR(1) | | | | | 0.000 |
| AR(2) | | | | | 0.171 |
| Hansen | | | | | 1 |
| N | 4318 | 4760 | 4760 | 4760 | 4760 |

Note: The values of t are given in brackets.

***, **, * represent the different significance levels (1%, 5%, 10% respectively). AR (1), AR (2) and the Hansen test report the p value corresponding to the statistics.

can find that after excluding government behaviors in provincial capital cities, ER can still promote the improvement of green technological innovation at the 5% significance level.

Second, most scholars regard green patent application and green patent authorization as ideal indicators for measuring green innovation. The number of green patent applications are a relevant indicator for approximating green technological innovation, but there may be relatively lagging. The number of green patents granted is a direct reflection of green innovation capabilities [86]. As result, the present study further uses the number of green patents granted as dependent variable to research the impact of ER on green technological innovation. We find from column (2) that, after replacing the green innovation capabilities (R-GTI), ER can still facilitate green technological innovation at the 5% significance level.

Thirdly, industrial solid waste comprehensive utilization rate and urban domestic sewage treatment rate were utilizing to characterize the ER instead of the comprehensive index [47], and the regression results are reported in column (3). The main variable regression coefficient symbol are consistent with the data above at the 1% significance level.

Additionally, In order to test the lagged and dynamic effects of environmental regulation on urban green technology innovation, we further construct models (4) and (5). The results in Column (4) tests the impact of one-period lagged environmental regulation on green technology innovation. We can see that one-period-lagged environmental regulation has a significant role in promoting green technology innovation in city. Considering that GTI may have dynamic inertia, the lagged term of the explained variable is used as the explanatory variable. Meanwhile, the GMM estimation method is used to test the model, and the specific results are shown in Column (5). From the results of the correlation test of the first-order AR (1) and second-order AR (2) sequences, it can be seen that there is no second order sequence autocorrelation, which shows that the model setting is reasonable. The p values of the Hansen test for the validity of the tool variables are all greater than 0.1, which accepts the original assumption that the tool variables do not have excessive constraints. The main variable regression coefficient symbol are consistent with the data above at the 1% significance level. In general, all robustness tests show that the empirical results of this paper are highly reliable.

## 4.3. Heterogeneity analysis

This paper also conducted a heterogeneous analysis of two aspects of the impact of ER on green technological innovation (GTI) in Table 5. First, the notable differences in regional economic development and geographical location across the 280 cities in China were divided in accordance with the eastern, central, and west pattern classification [87]. We separately examined the impact of ER in the eastern, central, and western regions on GTI. We find from columns (1) to (3) of Table 5 that ER in the central region can have a significant impact on GTI at the 5% significance level, while ER in the eastern and western regions has no significant impact on GTI. There may be two reasons: On the one hand, the economic development of western cities lags behind that of eastern and central cities, and economic growth is still the main goal of the western region construction. As a result, the environmental regulation in the western region has an insignificant role in promoting green technological innovation [95]. On the other hand, the economic development of the eastern region is far ahead of other regions [96], which has the basic conditions for environmental regulation to promote green technology innovation. However, there are differences in the degree of strictness of environmental regulation in different regions. Strict environmental regulation has caused the transfer of polluting enterprises in the eastern region, and has not produced a significant innovation promotion effect.

In addition, we examine the heterogeneity between resource-based cities (According to the plan for the sustainable development of resource-based cities in China (2013–2020) issued by

**Table 5. Heterogeneity test.**

| Variables | Area | | | City type | |
|---|---|---|---|---|---|
| | East | Central | Western | Resource-based | Non-resource |
| | (1) | (2) | (3) | (4) | (5) |
| ER | 0.330 | 0.496** | 0.131 | 0.650*** | 0.014 |
| | (1.36) | (2.12) | (0.58) | (3.24) | (0.07) |
| Control variables | YES | YES | YES | YES | YES |
| Constant | -17.83*** | -9.53*** | -4.33*** | -8.98*** | -11.15*** |
| | (-4.72) | (-2.79) | (-1.12) | (-2.84) | (-3.79) |
| City FE | YES | YES | YES | YES | YES |
| Year FE | YES | YES | YES | YES | YES |
| Adjusted $R^2$ | 0.670 | 0.548 | 0.482 | 0.612 | 0.581 |
| N | 1887 | 1853 | 1020 | 1921 | 2839 |

Note: The values of t are given in brackets.

***, **, * represent the different significance levels (1%, 5%, 10% respectively).

the State Council: resource-based cities are cities in which the mining and processing of natural resources such as minerals and forests in the region are the leading industries (including prefecture-level cities, districts and other county-level administrative districts, etc.). There are a total of 262 resource-based cities, including 126 prefecture-level administrative regions, 62 county-level cities, 58 counties, and 16 municipal districts. Available online at: http://www.gov.cn/zwgk/2013-12/03/content _2540 070.htm) and non-resource-based cities. Resource based cities mainly rely on the development of local resources to achieve economic development. The impact of ER on green technological innovation (GTI) may vary with city types. Columns (4) and (5) report the estimated results. We find that in resource-based cities, ER can indeed facilitate the increase of GTI at the 1% significance level, while this facilitation effect is not significant in non-resource-based cities. This show that the effectiveness of environmental regulations in resource-based areas in improving the efficiency of the green technological innovation is even more pronounced, which validates H2. In recent years, the state has continuously promoted green development, forcing local governments to strengthen environmental regulations and increase investment in scientific research, especially resource-based cities, thereby promoting the improvement of the level of green technology innovation in resource-based cities [93].

## 4.4. Mechanism analysis

This paper investigated the two mechanisms of ER affecting green technological innovation (GTI), and Table 6 reports the estimated results. Columns (1) to (2) test the mediating effect of foreign direct investment (FDI). The results show the coefficient of ER in columns (1) is 0.529 at the 1% significance level and the coefficient of FDI in columns (2) is 0.028 at 1% significance level, which indicates that the indirect effect is significant. Meanwhile, the coefficient of ER in columns (2) is 0.345 at 5% significance level, which indicating that the direct effect is significant. This shows that FDI has a mediating effect between ER and green technology innovation, and the indirect utility of ER through FDI to improve the level of GTI is 0.015, which validates H3. The reason is that those strict environmental regulations can attract environmentally friendly FDI and enhance the diffusion of cleaner production technologies [97, 98]. Meanwhile, FDI helps to promote green technology innovation in cities via the technology spillover effect.

**Table 6. Mechanism test.**

| Variables | FDI | GTI | GOT | GTI |
|---|---|---|---|---|
| | (1) | (2) | (3) | (4) |
| ER | 0.529** | 0.345** | 0.001** | 0.273** |
| | (2.59) | (2.58) | (2.03) | (2.07) |
| FDI | | 0.028** | | |
| | | (1.97) | | |
| GOT | | | | 46.98*** |
| | | | | (4.07) |
| Control variables | YES | YES | YES | YES |
| Constant | -1.951 | -19.26*** | -0.375*** | -8.883*** |
| | (-0.63) | (-11.83) | (-10.07) | (-3.98) |
| City FE | YES | YES | YES | YES |
| Year FE | YES | YES | YES | YES |
| Adjusted $R^2$ | 0.311 | 0.744 | 0.234 | 0.581 |
| N | 4760 | 4760 | 4760 | 4760 |

Note: The values of t are given in brackets.

***, **, * represent the different significance levels (1%, 5%, 10% respectively).

In addition, columns (3) to (4) of Table 6 tests the mediating effect of Government technology input (GOT). The results show the coefficient of ER in columns (3) is 0.001 at the 5% significance level and the coefficient of GOT in columns (4) is 46.98 at 1% significance level, which indicates that the indirect effect is significant. Meanwhile, the coefficient of ER in columns (2) is 0.273 at 5% significance level, which indicating that the direct effect is significant. This shows that the implementation of environmental regulations can promote the improvement of the level of green technology innovation through motivating GOT and the indirect utility of environmental regulation through GOT to improve the level of GTI is 0.12, which validates H4. When companies are under pressure from environmental regulations, local governments should actively adopt environmental subsidies, tax incentives and other policies to support the research and development of green technology innovation for companies [99]. This can let enterprises win the coordinated development of economic, social and environmental benefits, and form a win-win situation [100].

## 4.5. Discussion

Based on environmental regulation (ER) by the entropy method and urban green technology innovation (GTI) represented by the number of green patents, this study tested Porter's hypothesis (Porter and Van der Linde, 1991) at the level of Chinese prefecture-level cities, and the results of basic and theoretical analysis have been consistent. The overall impact of environmental regulations on urban green technology innovation is a significant role in promoting. This conclusion is consistent with the research of Chen et al. and in line with China's actual situation [55]. In the past ten years, in order to effectively reduce environmental pollution, the Chinese government has formulated and implemented a series of policies, such as establishing a national carbon emissions trading market, promoting law on cleaner production, and restricting the establishment of high-emission industries [15, 101]. Meanwhile, from 2014 to 2017, the number of green patent applications in China reached about 249,000, with an average annual growth rate of 3.7 percentage points higher than that of invention patents (State Intellectual Property Office of China, 2018).

In addition, due to the economic development and resource endowment of various places in China are large, it is necessary to analyze the impact of different regions and resource endowment environmental regulations on the innovation of green technology innovation. Heterogeneity test shows that the impact of environmental regulation on green technology innovation in central China and resource-based cities is more significant than that in eastern China, western China and non-resource-based cities. On the one hand, From the perspective of infrastructure, business environment and economic development level, the central region is ahead of the western region, but behind the eastern region, which is the best choice to undertake the industrial transfer of developed regions [102]. The higher industrial base and lower cost in the central region optimize the local innovation environment, which is conducive to the innovation and dissemination of green technologies [103]. On the other hand, Song et al. used the technical compensation theoretical framework using environmental regulation and found that environmental regulation "forcing" the green technology progress of resource-based enterprises [104]. Meanwhile, Facing the pressure of building resource -saving and environmentally friendly society, resource-based cities to promote technological progress with the "innovative compensation" effect of environmental regulation are important ways to achieve green development [105].

Finally, in order to explore the influence mechanism of environmental regulations on green technology innovation in the prefecture -level city, we used the intermediary effect model to analyze from FDI and government technology input (GOT). Through empirical research, foreign direct investment and government technological investment are two important ways for environmental regulations to promote green technology innovation. In the similar vein, the research of Muhammad and Khan found that effective environmental regulations can attract greener FDI from developed economies which will stimulate the development of green technological innovation in host countries [106]. Environmental regulation policy tools are an important threshold for foreign investment and introduction of advanced technologies. Reasonable environmental regulations can play a role in encouraging clean FDI inflows and limiting high pollution FDI inflows. In recent years, China's guidance to strengthen the inflow of FDI through environmental regulation policies has played an important role in promoting the innovation and development of green technology. Meanwhile, Acemoglu et al. theoretically found that the combination of government environmental pollution taxes and R&D subsidy policies can promote the development of clean technology [10]. This is due to the negative economic externality of GTI, which makes it difficult for profit-oriented enterprises to independently invest in GTI and other R&D [107]. Therefore, when issuing environmental policies and regulations, local governments need to give relevant enterprises appropriate funds for technological innovation research, so as to rapidly improve the level of GTI.

On the whole, the main contributions of this study are as follows. Firstly, as the world's largest developing country, the effect of China's environmental regulation policies often attracts much attention. We have found that environmental regulations have significantly promoted the innovation of green technology innovation in Chinese prefecture -level cities, which has important reference and reference significance for the implementation of environmental regulations in the world's developing countries. Secondly, we found that environmental regulation improves the level of green technology innovation through government technology investment and foreign direct investment, thereby promoting urban green development. This provides important reference opinions for the environmental policy makers of developing countries. Specifically, on the one hand, when formulating environmental regulations, policy makers should consider the impact of the introduction of environmental policies to attract environment -friendly foreign investment and reject the threshold of severe foreign investment. On the other hand, in order to better achieve the goal of green development, while

implementing increasingly stricter environmental regulations, the government needs to make appropriate financial subsidies on related enterprises. Additionally, this study differs from extant worked at geographical differences and ignored resource endowments, the conclusion of this paper not only enriches the heterogeneity research on the impact of environmental regulation on green technology innovation, but also provides a path reference for cities with different resource endowments to promote green technology innovation. Therefore, for policy makers, when implementing ER, it should pay attention to regional differences in regulatory execution and how to play ER's driving effect on GTI.

## 5. Conclusions and implications

### 5.1 Conclusions

There is no unified conclusion on the relationship between environmental regulation and green technology innovation in existing literature. Based on the data of 280 cities in China from 2003 to 2019, this paper adopted a double-fixed model to analyze the impact of environmental regulation on green technology innovation, and analyzed the impact of heterogeneous factors. Furthermore, this research identify the two transmission mechanisms of foreign direct investment and government technology input. The main conclusions are as follows.

First, environmental regulation improves the level of green technology innovation in Chinese cities, which indicates that the environmental policies and relevant environmental systems implemented by Chinese cities are effective and significantly promote urban green technology innovation. Moreover, this conclusion is still robust after the substitution of explanatory variables and explained variables and the deletion of some special samples.

Second, heterogeneity test results showed that in different economic geography environmental regulation of green technology innovation is different, the influence of environmental regulation of central China city of green technology innovation has significant improvement. But the eastern and western parts of China urban green technology innovation promoting effect is not significant. Under the condition of resources endowment difference, environmental regulation of resource-based cities with the resources city green technology innovation is different. The influence of the resource-based cities in environmental regulation for the promotion of green technology innovation effect significantly in the resources city, while the resources city environmental regulation for the promotion of green technology innovation effect is not obvious.

Third, by studying the mechanism of environmental regulation and green technology innovation, this paper finds that government technology input and foreign direct investment play a mediating role between environmental regulation and green technology innovation. It also shows that environmental regulation improves the level of green technology innovation in cities through government technology input and foreign direct investment.

### 5.2 Implications

The research conclusion of this paper has important reference and practical value for Chinese cities to implement "innovation-driven strategy" and "green development" policy, as well as provides experience for cities in most developing countries to improve the level of green technology innovation.

First, local governments should continue to improve and optimize environmental policies and strengthen environmental supervision of enterprises. On the one hand, strict environmental regulations will guide enterprises to conduct green technology research and development and accelerate the spread of green technology in the region. On the other hand, strict environmental regulations will crowd out polluting industries and accelerate the transformation of urban industrial structure to green [108].

Second, heterogeneity test shows that the regional economic difference and resource property difference lead to the influence of environmental regulation on green technology innovation there is a deviation. This requires that the central government to set up the environmental indicators for the local government environmental policy scope, promote the regional environmental cooperation, to avoid the transfer of "one size fits all" policy and pollution [62]. Therefore, local governments should scientifically formulate corresponding environmental policies according to the local environment and economic level, so as to ensure effective implementation of environmental policies [109, 110].

Thirdly, the improvement of urban green technology level not only needs to strengthen government technology input, but also needs to attract green technology, green production equipment and green management experience through foreign businesses. On the one hand, local governments should strengthen financial support for R&D of green technology innovation, such as green innovation subsidies and green loans, to reduce the cost pressure of green innovation and improve the level of green technology innovation of enterprises. On the other hand, local governments should create a good business environment to attract investment from advanced green industries and enterprises in developed countries and accelerate the process of urban green innovation [68]. Especially for some developing countries, it is necessary to pay attention not only to the differences of regional urban development, but also to the important role of government technology input and foreign direct investment in making environmental policies to promote green technology innovation.

## 5.3 Limitations

Although the results presented in this paper are robust and meaningful, the study in this paper still has the following limitations. First, due to the unobservability and unavailability of the data, the green patent data only represents the GTI level at the patent output level of the innovative agent, but not the true GTI level of the innovative agent. In the future, we should try to measure the true level of urban green technology innovation in multiple dimensions. Second, in this paper, we study the impact of environmental regulation on GTI in terms of intensity levels. In the future, the impact of different types of environmental regulation on green technology innovation can be studied.

## Supporting information

**S1 Data.**
(XLSX)

## Author Contributions

**Supervision:** Wen Gao.

**Writing – original draft:** Ming Zhang.

**Writing – review & editing:** Wancheng Xie.

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
