## [Decision Letter · Decision Letter 0]

8 Sep 2022

PONE-D-22-16530Have environmental regulations promoted green technological innovation in cities? Evidence from China's green patentsPLOS ONE

Dear Dr. Xie,

Thank you for submitting your manuscript to PLOS ONE. After careful consideration, we feel that it has merit but does not fully meet PLOS ONE’s publication criteria as it currently stands. Therefore, we invite you to submit a revised version of the manuscript that addresses the points raised during the review process.

We look forward to receiving your revised manuscript.

Kind regards,

Xingwei Li, Ph.D.

Academic Editor

PLOS ONE

Journal Requirements:

"NO"

4. Please ensure that you include a title page within your main document. You should list all authors and all affiliations as per our author instructions and clearly indicate the corresponding author.

5. We note you have included a table to which you do not refer in the text of your manuscript. Please ensure that you refer to Table 1 in your text; if accepted, production will need this reference to link the reader to the Table.

Additional Editor Comments:

I recommend that authors review the latest literature in the field of green technology innovation published in 2022.

e.g.,

Gharib, A. M., Palmer, M., & Zhang, M. (2022). Maintaining legitimacy: an institutional cooptative analysis of a green technology innovation scheme crisis. Innovation, 1-31.

Losacker, S. (2022). ‘License to green’: Regional patent licensing networks and green technology diffusion in China. Technological Forecasting and Social Change, 175, 121336.

Li, X., Dai, J., He, J., Li, J., Huang, Y., Liu, X., & Shen, Q. (2022). Mechanism of Enterprise Green Innovation Behavior Considering Coevolution Theory. International Journal of Environmental Research and Public Health, 19(16), 10453.

Li, X., Huang, Y., Li, J., Liu, X., He, J., & Dai, J. (2022). The mechanism of influencing green technology innovation behavior: evidence from Chinese construction enterprises. Buildings, 12(2), 237.

Li, X., Huang, Y., Li, X., Liu, X., Li, J., He, J., & Dai, J. (2022). How does the Belt and Road policy affect the level of green development? A quasi-natural experimental study considering the CO2 emission intensity of construction enterprises. Humanities and Social Sciences Communications, 9(1), 1-11.

etc.

In addition, I recommend that the authors carefully revise this manuscript based on the comments of the reviewers.

Reviewers' comments:

Reviewer's Responses to Questions

**Comments to the Author**

1. Is the manuscript technically sound, and do the data support the conclusions?

Reviewer #1: Yes

Reviewer #2: Yes

Reviewer #3: Yes

2. Has the statistical analysis been performed appropriately and rigorously? 

Reviewer #1: Yes

Reviewer #2: Yes

Reviewer #3: No

3. Have the authors made all data underlying the findings in their manuscript fully available?

Reviewer #1: Yes

Reviewer #2: No

Reviewer #3: Yes

4. Is the manuscript presented in an intelligible fashion and written in standard English?

Reviewer #1: Yes

Reviewer #2: No

Reviewer #3: Yes

5. Review Comments to the Author

Reviewer #1: First, the method used in this paper is relatively simple, but the overall process is relatively standard. Second, the statistical analysis process has not found problems. Third, it is known from the submission information that the authors have agreed to make the data public. Fourth, the English language is relatively clear, but there are some problems such as format in the manuscript.

Reviewer #2: This paper uses data from 280 Chinese cities from 2003 to 2019 to empirically answer the question whether environmental regulation can improve the level of urban green technology innovation. It is found that environmental regulation has a significant positive effect on green technology innovation. Under the heterogeneity of economic geographical region and resource-based city, environmental regulation has positive promoting effect on urban green technology innovation. Heterogeneity results also show that environmental regulation significantly promotes green technology innovation in central and resource-based cities, but does not significantly promote green technology innovation in eastern and western cities and non-resource-based cities. Further research shows that environmental regulation can promote the level of green technology innovation through the two transmission mechanisms of government technology input and foreign direct investment. For the Chinese local government which is implementing the policy of green economic transformation, the formulation of scientific regional environmental policy is beneficial to improve the level of green technology innovation, increase government technology input and optimize the foreign investment environment.

The aim of the analysis should be evidenced in the abstract and introduction sections. The methodology should be further explained for replication. Finally, the manuscript should be English proofread because some sentences are not clear.

Reviewer #3: This research topic of this paper is a classic problem, and scholars have carried out a lot of research. This paper uses data from 280 Chinese cities from 2003 to 2019 to empirically answer the question whether environmental regulation can improve the level of urban green technology innovation. However, compared with existing studies, the main problem is that the innovation is not prominent enough. Thus, this study needs to be elaborated and revised from the following aspects.

（1）The contributions and innovations of the article need to be further refined（line 29-44）. e.g. “First, compared with previous studies using provincial data or listed company data to analyze the impact of environmental regulation on technological innovation and green total factor productivity (Guo et al., 2017; Cao et al., 2019; Yasmeen et al., 2020)” . According to the literature review, we know there have been a lot of relevant studies from the city scale.

（2）There may be contradictions in the data description. e.g. “since the amount of industrial wastewater discharge will not be announced after 2010, this article draws on the research methods of Huang et al. (2018) using industrial waste water discharged, industrial sulfur dioxide emission, industrial soot(dust) emission, industrial solid waste comprehensive utilization rate and urban domestic sewage treatment rate.” I'm not sure why it should be used since the industrial wastewater discharge has not be announced after 2010?

（3）The analysis of empirical results and discussion need to strengthen the comparative analysis with existing research results, especially to highlight the particularity of the impact of environmental regulation on green innovation.

6. PLOS authors have the option to publish the peer review history of their article (what does this mean?). If published, this will include your full peer review and any attached files.

Reviewer #1: No

Reviewer #2: No

Reviewer #3: No

---

## [Author Response · Author response to Decision Letter 0]

2 Nov 2022

Dear Reviewers,

Thank you very much for your valuable comments and suggestions, which have played a great role in the revision and improvement of this paper. The following is the modification instructions we made according to your comments. At the same time, we also reviewed the paper and made some modifications and improvements. Please criticize and correct it again at your convenience.

1. Editor Comments:

 I recommend that authors review the latest literature in the field of green technology innovation published in 2022.

e.g.,

Gharib, A. M., Palmer, M., & Zhang, M. (2022). Maintaining legitimacy: an institutional cooptative analysis of a green technology innovation scheme crisis. Innovation, 1-31.

Losacker, S. (2022). ‘License to green’: Regional patent licensing networks and green technology diffusion in China. Technological Forecasting and Social Change, 175, 121336.

Li, X., Dai, J., He, J., Li, J., Huang, Y., Liu, X., & Shen, Q. (2022). Mechanism of Enterprise Green Innovation Behavior Considering Coevolution Theory. International Journal of Environmental Research and Public Health, 19(16), 10453.

Li, X., Huang, Y., Li, J., Liu, X., He, J., & Dai, J. (2022). The mechanism of influencing green technology innovation behavior: evidence from Chinese construction enterprises. Buildings, 12(2), 237.

Li, X., Huang, Y., Li, X., Liu, X., Li, J., He, J., & Dai, J. (2022). How does the Belt and Road policy affect the level of green development? A quasi-natural experimental study considering the CO2 emission intensity of construction enterprises. Humanities and Social Sciences Communications, 9(1), 1-11.

etc.

In addition, I recommend that the authors carefully revise this manuscript based on the comments of the reviewers.

The author's response: Thank you for your comments, which will help the manuscript track cutting-edge literature. We have added the latest literature closely related to the manuscript, including the literature mentioned above.

2. Reviewers' comments:

Reviewer #1:

 First, the method used in this paper is relatively simple, but the overall process is relatively standard. Second, the statistical analysis process has not found problems. Third, it is known from the submission information that the authors have agreed to make the data public. Fourth, the English language is relatively clear, but there are some problems such as format in the manuscript.

The author's response:

The author's response: Thank you for your opinion, which is very helpful for us to improve the manuscript. According to your comments, we adjusted and optimized the format of the manuscript.

Reviewer #2:

This paper uses data from 280 Chinese cities from 2003 to 2019 to empirically answer the question whether environmental regulation can improve the level of urban green technology innovation. It is found that environmental regulation has a significant positive effect on green technology innovation. Under the heterogeneity of economic geographical region and resource-based city, environmental regulation has positive promoting effect on urban green technology innovation. Heterogeneity results also show that environmental regulation significantly promotes green technology innovation in central and resource-based cities, but does not significantly promote green technology innovation in eastern and western cities and non-resource-based cities. Further research shows that environmental regulation can promote the level of green technology innovation through the two transmission mechanisms of government technology input and foreign direct investment. For the Chinese local government which is implementing the policy of green economic transformation, the formulation of scientific regional environmental policy is beneficial to improve the level of green technology innovation, increase government technology input and optimize the foreign investment environment.

The aim of the analysis should be evidenced in the abstract and introduction sections. The methodology should be further explained for replication. Finally, the manuscript should be English proofread because some sentences are not clear.

The author's response: Thank you for your advice, which is of great help to us in perfecting the manuscript. First, we add the purpose of the study in the abstract and introduction. Second, we further explain the methodology used in the manuscript. Finally, we proofread the language.

Reviewer #3:

This research topic of this paper is a classic problem, and scholars have carried out a lot of research. This paper uses data from 280 Chinese cities from 2003 to 2019 to empirically answer the question whether environmental regulation can improve the level of urban green technology innovation. However, compared with existing studies, the main problem is that the innovation is not prominent enough. Thus, this study needs to be elaborated and revised from the following aspects.

（1）The contributions and innovations of the article need to be further refined（line 29-44）. e.g. “First, compared with previous studies using provincial data or listed company data to analyze the impact of environmental regulation on technological innovation and green total factor productivity (Guo et al., 2017; Cao et al., 2019; Yasmeen et al., 2020)”. According to the literature review, we know there have been a lot of relevant studies from the city scale.

The author's response: Thank you for your suggestion, which has played an important role in helping us improve the manuscript. Based on your review comments, we re-revised and improved the contribution and innovation of the manuscript.

（2）There may be contradictions in the data description. e.g. “since the amount of industrial wastewater discharge will not be announced after 2010, this article draws on the research methods of Huang et al. (2018) using industrial waste water discharged, industrial sulfur dioxide emission, industrial soot(dust) emission, industrial solid waste comprehensive utilization rate and urban domestic sewage treatment rate.” I'm not sure why it should be used since the industrial wastewater discharge has not be announced after 2010?

The author's response: Thank you for your comments, which have played an important role in improving our manuscript. By consulting the relevant information, we get the following answer. According to the data of China City Statistical Yearbook, the industrial wastewater discharge standard ended in 2010, but the industrial wastewater discharge is updated every year. Therefore, the data of industrial wastewater discharge selected in this paper is continuous without year interruption.

The industrial wastewater discharge standard refers to the industrial wastewater discharge quantity that all the indexes have reached the national or local discharge standard. The industrial wastewater discharge refers to the amount of industrial wastewater discharged to the outside of the enterprise through all the discharge ports in the factory during the reporting period. 

（3）The analysis of empirical results and discussion need to strengthen the comparative analysis with existing research results, especially to highlight the particularity of the impact of environmental regulation on green innovation.

The author's response: Thank you for your comments, which have played an important role in improving our manuscript. According to your comments, we have added a new section (4.5) in the manuscript to discuss the empirical results. In the discussion section, we add a comparative analysis with the existing literature, highlighting the particularity of the impact of environmental regulation on green technology innovation.

3. Reviewer's Responses to Questions

Q1. The range of literature sources are inadequate. Specifically, it didn’t include most recent literature related to enhance the validity of the research questions and related argument. Therefore, it has ignored most recent literature from the related field.

A1: Special thanks to the editors and reviewers for their comments, this paper has made new revisions to the literature review of the manuscript. This paper adds the latest literature in the research field and improves the validity of the research questions and related arguments. The specific newly added literature is as follows:

1. Li, X., Dai, J., He, J., Li, J., Huang, Y., Liu, X., Shen, Q. (2022a). Mechanism of Enterprise Green Innovation Behavior Considering Coevolution Theory. International Journal of Environmental Research and Public Health, 19(16), 10453. Doi:10.3390/ijerph191610453.

2. Li, X., Huang, Y., Li, X., Liu, X., Li, J., He, J., Dai, J. (2022b). How does the Belt and Road policy affect the level of green development? A quasi-natural experimental study considering the CO2 emission intensity of construction enterprises. Humanities and Social Sciences Communications, 9(1), 1-11. Doi:10.1057/s41599-022-01292-4.

3. Li, X., Huang, Y., Li, J., Liu, X., He, J., Dai, J. (2022c). The mechanism of influencing green technology innovation behavior: evidence from Chinese construction enterprises. Buildings, 12(2), 237. Doi:10.3390/buildings12020237.

4. Losacker, S. (2022). ‘License to green’: Regional patent licensing networks and green technology diffusion in China. Technological Forecasting and Social Change, 175, 121336. Doi: 10.1016/j.techfore.2021.121336.

5. Fu, S., Ma, Z., Ni, B., Peng, J., Zhang, L., Fu, Q. (2021). Research on the spatial differences of pollution-intensive industry transfer under the environmental regulation in China. Ecological Indicators, 129, 107921. Doi:10.1016/j.ecolind.2021.107921.

Q2. In section 3.1, the explanation of the mediation effect test steps should refer to relevant literature.

A2: The paper is supplemented with relevant references in Section 3.1. This paper refers to the model of Baron and Kenny (1986) on mediating effect to test Hypothesis 3.

Baron, R. M., Kenny, D. A. (1986). The moderator–mediator variable distinction in social psychological research: Conceptual, strategic, and statistical considerations. Journal of personality and social psychology, 51(6), 1173. https://psycnet.apa.org/buy/1987-13085-001

Q3. The impact of environmental regulation on green technology innovation will take some time to complete. How to test the dynamic effect or lagged effect?

A3: In order to test the lagged and dynamic effects of environmental regulation on urban green technology innovation, we further construct models (4) and (5). The model (4) tests the impact of one-period lagged environmental regulation on green technology innovation. We can see that one-period-lagged environmental regulation has a significant role in promoting green technology innovation in city. Considering that GTI may have dynamic inertia, the lagged term of the explained variable is used as the explanatory variable. Meanwhile, the GMM estimation method is used to test the model, and the specific results are shown in Column (5). Meanwhile, the main variable regression coefficient symbol are consistent with the data above at the 1% significance level. In general, all robustness tests show that the empirical results of this paper are highly reliable.

Q4. In section 3.3, should explain the basis for data selection, that is, explain the reason for the deadline of data in 2019. If there are new policy regulations in the last two years, the research findings need to be discussed in relation to the latest policy.

A4: The missing values of China City Statistical Yearbook in 2020 are relatively serious, and the data of industrial sulfur dioxide emissions, industrial smoke and dust emissions, and industrial wastewater involved in this paper are missing. Meanwhile, due to the impact of the COVID-19 pandemic, the data for Chinese cities in 2020 fluctuated considerably compared to previous years. In addition, in 2020, many regions were shut down due to the COVID-19 pandemic, and the patent application time was also delayed compared with previous years. In general, this paper uses data from 2003 to 2019 by excluding the impact of missing data and COVID-19.

In addition, according to the data of China City Statistical Yearbook, the industrial wastewater discharge standard ended in 2010, but the industrial wastewater discharge index is updated every year. Therefore, the data of industrial wastewater discharge selected in this paper is continuous without year interruption.

Q5. This paper should create a discussion section that will answer the question of what the new article brings to science and how it contributes to the strengthening of green development. And the author should make comparison, contrast of previous paper and this paper to highlight the theoretical importance.

A5: We have added a discussion section to the article, which is detailed below.

Based on environmental regulation (ER) by the entropy method and urban green technology innovation (GTI) represented by the number of green patents, this study tested Porter's hypothesis (Porter and Van der Linde, 1991) at the level of Chinese prefecture-level cities, and the results of basic and theoretical analysis have been consistent. The overall impact of environmental regulations on urban green technology innovation is a significant role in promoting. This conclusion is consistent with the research of Chen (2018) and in line with China's actual situation. In the past ten years, in order to effectively reduce environmental pollution, the Chinese government has formulated and implemented a series of policies, such as establishing a national carbon emissions trading market, promoting law on cleaner production, and restricting the establishment of high-emission industries (Guo et al., 2017; Cheng et al., 2022). Meanwhile, from 2014 to 2017, the number of green patent applications in China reached about 249,000, with an average annual growth rate of 3.7 percentage points higher than that of invention patents (State Intellectual Property Office of China, 2018). 

In addition, due to the economic development and resource endowment of various places in China are large, it is necessary to analyze the impact of different regions and resource endowment environmental regulations on the innovation of green technology innovation. Heterogeneity test shows that the impact of environmental regulation on green technology innovation in central China and resource-based cities is more significant than that in eastern China, western China and non-resource-based cities. On the one hand, From the perspective of infrastructure, business environment and economic development level, the central region is ahead of the western region, but behind the eastern region, which is the best choice to undertake the industrial transfer of developed regions (Fu et al., 2021). The higher industrial base and lower cost in the central region optimize the local innovation environment, which is conducive to the innovation and dissemination of green technologies (Losacker, 2022). On the other hand, Song et al. (2022) used the technical compensation theoretical framework using environmental regulation and found that environmental regulation “forcing” the green technology progress of resource-based enterprises. Meanwhile, Facing the pressure of building resource -saving and environmentally friendly society, resource -based cities to promote technological progress with the "innovative compensation" effect of environmental regulation are important ways to achieve green development (Zhang and Qu 2020).

Finally, in order to explore the influence mechanism of environmental regulations on green technology innovation in the prefecture -level city, we used the intermediary effect model to analyze from FDI and government technology input (GOT). Through empirical research, foreign direct investment and government technological investment are two important ways for environmental regulations to promote green technology innovation. In the similar vein, the research of Muhammad and Khan (2019) found that effective environmental regulations can attract greener FDI from developed economies which will stimulate the development of green technological innovation in host countries. Environmental regulation policy tools are an important threshold for foreign investment and introduction of advanced technologies. Reasonable environmental regulations can play a role in encouraging clean FDI inflows and limiting high pollution FDI inflows. In recent years, China's guidance to strengthen the inflow of FDI through environmental regulation policies has played an important role in promoting the innovation and development of green technology. Meanwhile, Acemoglu et al. (2012) theoretically found that the combination of government environmental pollution taxes and R&D subsidy policies can promote the development of clean technology. This is due to the negative economic externality of GTI, which makes it difficult for profit-oriented enterprises to independently invest in GTI and other R&D (Shi et al., 2020). Therefore, when issuing environmental policies and regulations, local governments need to give relevant enterprises appropriate funds for technological innovation research, so as to rapidly improve the level of GTI.

On the whole, the main contributions of this study are as follows. Firstly, as the world's largest developing country, the effect of China's environmental regulation policies often attracts much attention. We have found that environmental regulations have significantly promoted the innovation of green technology innovation in Chinese prefecture -level cities, which has important reference and reference significance for the implementation of environmental regulations in the world's developing countries. Secondly, we found that environmental regulation improves the level of green technology innovation through government technology investment and foreign direct investment, thereby promoting urban green development. This provides important reference opinions for the environmental policy makers of developing countries. Specifically, on the one hand, when formulating environmental regulations, policy makers should consider the impact of the introduction of environmental policies to attract environment -friendly foreign investment and reject the threshold of severe foreign investment. On the other hand, in order to better achieve the goal of green development, while implementing increasingly stricter environmental regulations, the government needs to make appropriate financial subsidies on related enterprises. Additionally, this study differs from extant worked at geographical differences and ignored resource endowments, the conclusion of this paper not only enriches the heterogeneity research on the impact of environmental regulation on green technology innovation, but also provides a path reference for cities with different resource endowments to promote green technology innovation. Therefore, for policy makers, when implementing ER, it should pay attention to regional differences in regulatory execution and how to play ER's driving effect on GTI.

Q6. In conclusion, this paper should indicate the directions of further research and research limitations.

A6: This paper adds limitations in the last section of Chapter 5 of the manuscript.

Although the results presented in this paper are robust and meaningful, the study in this paper still has the following limitations. First, due to the unobservability and unavailability of the data, the green patent data only represents the GTI level at the patent output level of the innovative agent, but not the true GTI level of the innovative agent. In the future, we should try to measure the true level of urban green technology innovation in multiple dimensions. Second, in this paper, we study the impact of environmental regulation on GTI in terms of intensity levels. In the future, the impact of different types of environmental regulation on green technology innovation can be studied.

---

## [Decision Letter · Decision Letter 1]

28 Nov 2022

Have environmental regulations promoted green technological innovation in cities? Evidence from China's green patents

PONE-D-22-16530R1

Dear Dr. Xie,

We’re pleased to inform you that your manuscript has been judged scientifically suitable for publication and will be formally accepted for publication once it meets all outstanding technical requirements.

Kind regards,

Xingwei Li, Ph.D.

Academic Editor

PLOS ONE

Additional Editor Comments (optional):

Reviewers' comments:

Reviewer's Responses to Questions

**Comments to the Author**

1. If the authors have adequately addressed your comments raised in a previous round of review and you feel that this manuscript is now acceptable for publication, you may indicate that here to bypass the “Comments to the Author” section, enter your conflict of interest statement in the “Confidential to Editor” section, and submit your "Accept" recommendation.

Reviewer #1: All comments have been addressed

Reviewer #2: All comments have been addressed

2. Is the manuscript technically sound, and do the data support the conclusions?

Reviewer #1: Partly

Reviewer #2: Yes

3. Has the statistical analysis been performed appropriately and rigorously? 

Reviewer #1: Yes

Reviewer #2: Yes

4. Have the authors made all data underlying the findings in their manuscript fully available?

Reviewer #1: Yes

Reviewer #2: Yes

5. Is the manuscript presented in an intelligible fashion and written in standard English?

Reviewer #1: Yes

Reviewer #2: Yes

6. Review Comments to the Author

Reviewer #1: The authors have made great improvements in literature, empirical methods, discussion and other parts, and the revised paper is more complete.

Reviewer #2: The paper has been improved according to the reviewers' comments. Now the manuscript can be accepted for publication.

7. PLOS authors have the option to publish the peer review history of their article (what does this mean?). If published, this will include your full peer review and any attached files.

Reviewer #1: No

Reviewer #2: No

---

## [Editor Report · Acceptance letter]

2 Dec 2022

PONE-D-22-16530R1 

Have environmental regulations promoted green technological innovation in cities? Evidence from China's green patents 

Dear Dr. Xie:

I'm pleased to inform you that your manuscript has been deemed suitable for publication in PLOS ONE. Congratulations! Your manuscript is now with our production department. 

Kind regards, 

on behalf of

Prof. Dr. Xingwei Li 

Academic Editor

PLOS ONE